# In Vitro Effects of Waterborne Polyurethane 3D Scaffolds Containing Poly(lactic-co-glycolic acid)s of Different Lactic Acid/Glycolic Acid Ratios on the Inflammatory Response

**DOI:** 10.3390/polym15071786

**Published:** 2023-04-04

**Authors:** Guanyu Zhang, Ao Zhen, Jinlin Chen, Bohong Du, Feng Luo, Jiehua Li, Hong Tan

**Affiliations:** State Key Laboratory of Polymer Materials Engineering, College of Polymer Science and Engineering, Sichuan University, Chengdu 610065, China

**Keywords:** degradation velocity, inflammatory response, microglial cells, polyurethane, scaffold

## Abstract

The physical and chemical properties of tissue engineering scaffolds have considerable effects on the inflammatory response at the implant site in soft tissue repair. The development of inflammation-modulating polymer scaffolds for soft tissue repair is attracting increasing attention. In this study, in order to regulate the inflammatory response at the implant site, a series of waterborne polyurethane (WPU) scaffolds with different properties were synthesized using polyethylene glycol (PEG), polycaprolactone (PCL) and poly (lactic acid)–glycolic acid copolymers (PLGAs) with three lactic acid/glycolic acid (LA/GA) ratios as the soft segments. Then, scaffolds were obtained using freeze-drying. The WPU scaffolds exhibited a porous cellular structure, high porosity, proper mechanical properties for repairing nerve tissue and an adjustable degradation rate. In vitro cellular experiments showed that the degradation solution possessed high biocompatibility. The in vitro inflammatory response of C57BL/6 mouse brain microglia (immortalized) (BV2) cells demonstrated that the LA/GA ratio of the PLGA in WPU scaffolds can regulate the external inflammatory response by altering the secretion of IL-10 and TNF-α. Even the IL-10/TNF-α of PU5050 (3.64) reached 69 times that of the control group (0.053). The results of the PC12 culture on the scaffolds showed that the scaffolds had positive effects on the growth, proliferation and differentiation of nerve cells and could even promote the formation of synapses. Overall, these scaffolds, particularly the PU5050, indeed prevent BV2 cells from differentiating into a pro-inflammatory M1 phenotype, which makes them promising candidates for reducing the inflammatory response and repairing nerve tissue. Furthermore, PU5050 had the best effect on preventing the transformation of BV2 cells into the pro-inflammatory M1 phenotype.

## 1. Introduction

Tissue engineering (TE) is an emerging field that applies high-performance biomaterials to damaged tissue with the aim of to maintaining, restoring and improving tissue function [1]. As biological materials continue to develop, the 3D porous scaffolds implanted into damaged sites for tissue repair are receiving increasing attention [2]. These scaffolds act as an extracellular matrix (ECM) in the damaged site, providing a fulcrum for cell adhesion [3] and transporting metabolic waste [4] and nutrients [5] to accelerate tissue repair. In recent years, important progress has been made in tissue engineering scaffolds for the repair and regeneration of myocardial [6], bone [7], nerve [8], skin [9] and other tissues [10]. The repairing effect of 3D porous scaffolds on damaged tissues depends on the properties of the scaffolds themselves [11]. The scaffold’s properties, such as its topological structure [12], pore structure [13], hemolysis rate [14], mechanical properties [15] and degradation rate [16], are critical factors in tissue repair. Sun [17] et al. reviewed advances in three-dimensional nanofibrous macrostructures using electrospinning and summarized the effects of microstructures’ material characteristics, such as pore structure, porosity and pore size, on tissue repair. Saad et al. [18] designed a porous magnesium scaffold to repair human cancellous bone and explored the influence of flow rates on the scaffold’s dynamic degradation behavior. Lin et al. [19] created degradable WPU scaffolds with an aligned three-dimensional pore structure using freeze-drying, and they were able to promote cell adhesion and tissue repair.

Many materials have been applied to the synthesis of tissue engineering scaffolds including natural and synthetic materials [20]. The most-used natural materials for this purpose include gelatin [21], collagen [22], sodium alginate [23], etc., and the most-used synthetic biomaterials include polycaprolactone (PCL) [24], polycarbonate diol (PCDL) [25], polyurethane (PU) [26], etc. Among these materials, polyurethane, which has gained significant attention due to its good biocompatibility and adjustability, is a versatile biomaterial [27]. By adjusting the proportion of hard and soft segments, the properties of PU that influence tissue regeneration performance, such as its degradation rate [28], mechanical properties [29], 3D structure [30], hydrophilicity [31] and crystallinity [32], can easily satisfy the needs of different tissues. Therefore, biodegradable polyurethane scaffolds are used as soft tissue scaffolds to repair various tissue organizations (e.g., myocardial [33], nerves [34] and angiogenesis [35]). For example, Hun et al. [36] created a 3D-printed polyurethane ink by mixing bioactive ingredients (TGFβ3 or the small-molecule drug Y27632) with biodegradable polyurethane (PU) elastic nanoparticles to successfully repair rabbit cartilage. In a study by Niu et al. [37], the authors prepared a tissue-engineered scaffold for multilayered biological interfaces based on alternate-segment polyurethane. This is a layered-nanofiber tubular scaffold that promotes the cell adhesion and vascularization of new tissue. These studies demonstrate that biodegradable polyurethane scaffolds have outstanding biocompatibility and have expanded the application of PU in tissue engineering.

Waterborne polyurethane (WPU) uses water as a solvent to disperse nano-scale polyurethane particles, fundamentally solving the problem of volatile organic compounds [38], and can easily be used to fabricate 3D porous scaffolds with freeze-drying [39]. Many researchers recognize the eminent potential of WPU; for example, Shan-hui Hsu and colleagues have made remarkable contributions to this field through their exploration of the applications of WPU in repairing nerve tissue [40], cell therapy vectors [41] and antibacterial coatings [42]. Our research group has also undertaken a variety of studies on the application of waterborne polyurethane scaffolds in soft tissue repair. For example, WPU nerve scaffolds with adjustable mechanical properties were designed, and we found that the modulus of a scaffold has a great influence on the regeneration of nerve tissue [43]. Additionally, an injectable WPU hydrogel for vocal cord repair [44] and a WPU double-layer mesh for periodontal tissue repair [45] have been developed, and a WPU scaffold with sustained drug release for the in situ targeted treatment of cancer has been investigated with in vitro and in vivo experiments [46]. The injury of nervous tissue is a common disease in neurosurgery. According to current statistics, the number of new nerve tissue injuries in the world each year reaches 50–60 million, among which severe nerve injuries account for 10% [47]. Therefore, the search for a new type of suitable materials to repair damaged nerve tissue is one of the key points of neurosurgery. The wet-state modulus of WPU scaffolds can be easily adjusted to 2–200 Kpa, which is a very ideal condition for nerve tissue repair. Thus, WPU 3D porous scaffolds are often used to repair nerve tissue. Chen et al. [48] mixed WPU with Bletilla striata polysaccharide to make a bio-ink and electrospun a 3D scaffold for nerve regeneration and scar prevention. Hsu et al. [49] made a novel flexible nerve conduit using waterborne biodegradable polyurethane for peripheral nerve regeneration.

With the deepening of research on this topic, researchers are increasingly finding that the inflammatory response is not solely an obstacle in the process of tissue repair. On the contrary, it is a necessary process and plays a positive role in tissue repair [50]. Chemokines, metabolites, growth factors and immune cells all play non-substitutable roles in this process [51]. Among these factors, the change in the immune microenvironment caused by the differentiation of immune cells into different phenotypes is more significant [52]. In general, immune cells can differentiate into anti-inflammatory and pro-inflammatory phenotypes, and the ratio of these two cells has a great influence on the immune microenvironment, which, in turn, influences the effectiveness of tissue repair [53]. Therefore, exploring the influence of scaffolds on the differentiation of immune cells is of great significance for tissue repair. Some researchers use additional chemokines [54] and immune cells [55] to reduce the inflammatory response at the implantation site, while others focus on the properties of the scaffold itself such as topology [56], modulus [57] and pore structure [58]. In addition, some segments such as PLGA are often used in tissue engineering [59]. Many researchers applied PLGA to repair bone tissue [60], osteochondral [61] and nerve tissue [62]. In addition, researchers have noted that the acidic environment produced by the degradation of PLGA or PLA components has a great impact on the immune environment at the implantation site [63]. Nevertheless the appropriate PLA or PLGA content can reduce inflammation, and our previous studies have also shown that by introducing proper PLGA into the chain segment of waterborne polyurethane, the obtained scaffold can promote the transformation of microglia from M1 to M2, which is conducive to the growth of the axons of nerve cells PC12 [64]. However, the regulation of immune response by changing LA/GA of PLGAs in WPU and then using such a WPU to repair nerve tissue has not been explored.

In this work, the percentage of PLGA in the soft segment was fixed at 20%, and a series of WPUs were prepared to explore the effect of the LA/GA ratio in soft segments on the inflammatory response of BV2 cells. The soft segment of WPUs consisted of PEG-1450, PCL-2000 and three kinds of PLGA-2000 containing different LA/GA ratios (LA/GA = 50:50, 75:25 and 90:10), while the hard segment consisted of LDI, PDO and L-lysine. The three different WPUs were labeled as PU9010, PU7525 and PU5050. Then, the emulsion was freeze-dried to obtain 3D porous scaffolds. Following this, the properties of the scaffolds were characterized, such as their degradation performance, mechanical properties, water absorption and pore structure, and the influence of changes in these properties on the inflammatory response of BV2 microglia cells was observed in vitro. Finally, PC12 cells were cultured on WPU scaffolds to verify that the WPU scaffolds could support the growth of nerve cells. In summary, our work synthesized waterborne polyurethanes using three kinds of PLGA and freeze-dried them into scaffolds, and also modulated the BV2 cell differentiation phenotype and reduced the immune reaction by changing the LA/GA ratio of the PLGAs. The experimental design is shown in Figure 1.

## 2. Materials and Methods

### 2.1. Materials

Polyethylene glycol (PEG, molecular weight = 1450, Dow Chemical Company, Midland, MI, USA) and polycaprolactone (PCL, molecular weight = 2000, Daicel Investment Co., Ltd., Shanghai, China) were dehydrated at 90–100 °C under vacuum for 1.5 h. L-Lysine Diisocyanate (LDI) and 1,3-propanediol (PDO) were redistilled under vacuum before use. Poly (lactic-co-glycolic acid) (PLGA, molecular weight = 2000, LA/GA ratio = 50:50, 75:25, 90:10) was purchased from Ruixi Biotechnology Co., Ltd., Xi’an, China. L-Lysine, used as a chain extender, was purchased from Emei Mountain Ronggao Biochemical Products Co., Ltd., Chengdu, China. PDO was purchased from TCI Chemical Industry Development Co. Ltd., Shanghai, China.

### 2.2. WPU Synthesis

The waterborne polyurethane emulsion was obtained using a typical acetone method. Polyols of PCL and PEG, and the PLGA, were added into a three-neck round-bottom flask equipped with an overhead propeller and a thermometer. Next, PEG, PCL and the PLGA were dehydrated for 1.5 h under vacuum at 95 °C; meanwhile, we pumped N_2_ into the flask three times (pumping N_2_ every 0.5 h). Then, LDI and 0.1% organic bismuth (a catalyst) were added to the flask, under the protection of nitrogen, and prepolymerized for 2.5 h at 90 °C. Following this, we reduced the reaction temperature to below 60 °C. Next, PDO was added to the flask, and the WPU chain was extended for 1 h at 68 °C. When the chain extender reaction was complete, acetone was added into the system to reduce the viscosity of the prepolymer. The L-lysine solution was then added drop-by-drop and reacted for 5 min. Finally, water was added to the reaction system, and at the same time, the rotation speed of the overhead propeller was set to 1500 r/min. After 1.5 h of intense stirring and defoaming, waterborne polyurethane emulsions were obtained. The compositions of our WPUs are listed in Table 1. These WPUs are labeled as PU x: y, which means the LA/GA ratio of the PLGA 2000 is x: y. The WPU synthesis process is shown in Figure 2.

### 2.3. Preparation of WPU Scaffolds with Three LA/GA Ratios

Firstly, the solid content of the WPU emulsion was adjusted to 14% using a rotary evaporator. Then, the adjusted emulsion was injected into a mold using a pipetting gun. We placed the mold in a 4 °C environment for 4 h, and then in a −20 °C environment for 24 h, and, finally, in a freeze-dryer (Pilot10-15M, Biocool, Beijing, China) for 24 h. The freeze-dried scaffolds were stored in a thermostatic air-drying oven [65].

### 2.4. Characterization Analysis of WPU Scaffolds with Three LA/GA Ratios

#### 2.4.1. Fourier Transform Infrared (FTIR) Spectroscopy

The waterborne polyurethane film samples were characterized using a Nicolet-6700 spectrometer with an attenuated total reflectance (ATR) objective and equipped with a zinc selenide crystal (iS50 FT-IR, NICOLET, Thermo Scientific, Waltham, MA, USA). The spectra were collected at a spectral resolution of 4 cm^−1^ by accumulating 32 scans [66].

#### 2.4.2. Microstructure

The prepared scaffolds were brittle-broken in liquid nitrogen. After spraying gold on their surfaces, the scaffolds were characterized using scanning electron microscopy (SEM) (Nova NanosM 450, FEI Company, Hillsboro, OR, USA) [67].

Similarly, scaffolds with different degradation times were brittle-broken in liquid nitrogen. Cross-sections and the surfaces of scaffolds were observed using scanning electron microscopy after degradation to explore changes in their internal and external surface morphology.

#### 2.4.3. Porosity and Pore Size

Image J was used to analyze the SEM images of the WPU scaffolds, and then we calculated the average pore size. The porosity of the scaffolds was calculated using the following formula [64]:(1)$$Porosity=100\%*\frac{V_{Scaffold}*\rho_{PU}-m_{scaffold}}{V_{Scaffold}*\rho_{PU}}$$
where $V_{Scaffold}$ represents the volume of the scaffolds (obtained with measurements), $\rho_{PU}$ represents the density of the scaffolds (obtained with calculation) and $m_{scaffold}$ represents the mass of the scaffolds (obtained with measurements).

#### 2.4.4. Water Absorption

The WPU scaffolds with three LA/GA ratios were shaped into uniform cuboids; the weights of the cuboids and the dry samples were labeled as *W_d_*. Then, the cuboids were added to EP tubes filled with deionized water, and the EP tubes were placed in a shaker at 37 °C. After 24 h, the scaffolds were taken out. Accordingly, the weight of the wet samples was recorded as *W_d_*. The following formula was used to calculate the water absorption of the scaffolds:(2)$$WA\%=\frac{W_{w}-W_{d}}{W_{d}}\times 100\%$$

#### 2.4.5. Mechanical Properties

The mechanical properties of the scaffolds, in both dry and wet states, were characterized using the universal tensile tester (HZ-1004, Lixian Instrument Scientific Co., Ltd., Dongguan, China). The compression speed was set at 10 mm/min, and a 1 Kg mechanical sensor was used. In the compression process, the test would stop when the maximum deformation reached 70%. All the scaffolds were characterized at 25 °C.

#### 2.4.6. Degradation Properties

Weight retention ratio. The uniform dry cuboid WPU scaffolds were added to EP tubes after weighting, and the masses of the dry scaffolds were recorded as $W_{pre}$. Next, we poured moderate PBS solution into the tubes to submerge the scaffolds. Then, the tubes were placed into a shaker (the rotation rate was 90 rpm, and the temperature was 37 °C). When the preset time was reached, the scaffolds were taken out and placed in a −20 °C environment for 12 h after being washed 4 times using deionized water. Then, the WPU scaffolds were placed in a freeze-dryer. After freeze-drying, the WPU scaffolds were weighed, and their masses were labeled as $W_{pro}$. The following formula was used to calculate the weight retention ratio of the WPU scaffolds:(3)$$WR\%=\frac{W_{pro}}{W_{pre}}\times 100\%$$

The changes in microstructure: The microstructure of each scaffold at every scheduled time stage in the in vitro degradation process was characterized using SEM (Nova NanoSEM450 at 5 kV).

#### 2.4.7. Water Contact Angle

The hydrophilicity of the WPU films was characterized using a video-based optical contact measuring instrument (DSA100, KRUSS, Shanghai, China) at a room temperature of 25 °C. During the characterization process, the water contact angle was measured at 60 s (n ≥ 3) after the droplet was dropped on the material.

### 2.5. Cell Experiment

#### 2.5.1. Cell Culture

L929 cells were provided by the Key Laboratory of Transplantation and Immunity, Ministry of Education, West China Science and Technology Park, Sichuan University. The cells were cultured in complete Dulbecco’s Modified Eagle Medium (DMEM, Gibco Life, Grand Island, NY, USA) supplemented with 10% fetal bovine serum (FBS, Hyclone, Logan, UT, USA), 100 units/mL penicillin and 100 units/mL streptomycin (Gibco) at 37 °C.

BV2 cells and PC12 cells were purchased from Huaan biological company and Sari biological company, respectively. First, the materials were sent to the Sichuan Institute of Nuclear Energy for γ-ray irradiation sterilization. Then, the scaffolds were soaked in PBS overnight in an incubator. Before inoculating the cells, the material was placed in a 48-well plate. The material was cleaned twice using PBS and the complete medium, respectively, for 5 min each time. All three types of cells were cultured in a 5% CO_2_ incubator, the temperature of which was set at 37 °C. The cells were digested in the culture flask using trypsin and terminated by adding complete culture medium before use. Following this, we centrifuged the cells for 4 min at 1500 r/min, poured out the supernatant and added fresh complete culture medium before counting the cells numbers. We then added 50,000 cells/well and 1 mL medium/well to a 48-well plate with the material; the culture medium of PC12 cells contained 50 ng/mL NGF. After being cultured 24 h to induce adhesion, the culture medium of the BV2 cells was replaced with complete culture medium containing lipopolysaccharide (LPS) or no LPS. When the preset time was reached, the medium was sucked out, and the material was cleaned twice using sterilized PBS to remove the residual medium. The cells were cultured for 1 and 6 days, respectively.

#### 2.5.2. Biocompatibility of Degradation Solution

Degradation solutions with different degradation times were obtained using hydrolysis. First, the degradation solution (at 1, 2 and 5 weeks) was filtered and disinfected using a 0.22 μm filter. Second, 10%, 1% and 1‰ diluted degradation solutions were obtained using a serum-containing cell culture medium. The biotoxicity of the scaffold degradation solution was characterized using the MTT method. L929 cells were digested, centrifuged and added to the culture medium to obtain a cell suspension. Then, we used red blood cell counter plates to estimate the cell concentration in the suspension. At a cell density of 5 × 10^3^ cells per well, the cells were seeded at the bottom of a 96-well plate and cultured in an incubator at 37 °C for 24 h. We sucked out the original medium from each well and adding 150 μL diluted medium containing the degradation solution. After culturing for a certain amount of time, the relative cell number was determined using the MTT method. The O.D. measurements of the cells in the degradation solution were divided by the measurements from the control sample to obtain the survival percentage. We used three parallel samples for each concentration of the degradation solution. The cells in the control group were cultured using a medium without a degradation solution.

#### 2.5.3. NO Release Characterization

Each well of the 96-well plate was injected with 50 μL emulsion, and then WPUs were freeze-dried in cylindrical scaffolds 6 mm in diameter. We seeded 7 × 10^4^ BV2 cells on WPU scaffolds for 24 h to induce cell adhesion, and then the WPU scaffolds were soaked in complete medium, with or without LPS, for 24 h. We removed 50 mL of the supernatant, and a BiYunTian nitric oxide kit was used to characterize the NO release of BV2.

#### 2.5.4. ELISA Kit Test

The concentrations of tumor necrosis factor TNF-α and anti-inflammatory factor IL-10 secreted by BV2 on the PU5050, PU7525 and PU9010 scaffolds were characterized using an ELISA kit. We then seeded 2 × 10^5^ BV2 cells on waterborne polyurethan scaffolds and tissue culture plates (TCPs). After the cells had been cultured according to the above method for the specified time, the medium supernatant was removed and characterized using the ELISA kit. 

#### 2.5.5. Western Blot (WB) Analysis of the BV2 Cells and PC12 Cells

The BV2 cells were seeded on the WUP scaffolds and cultured for 24 h. After adhesion, complete medium with or without LPS was added to replace the original medium (as in the above-mentioned ELISA and NO tests). Then, they were cultured for 24 h, and the BV2 cells were washed from the scaffolds. Using lysing, we collected the BV2 cells and extracted the total protein using a Protease Inhibitor Cocktail kit; then, a BCA (bicinchoninic acid) quantitative kit was used to quantify the total protein. For the PC12 cells, cell culture and protein extraction were also performed using the above method, except that they were not treated with LPS. The PC12 cells were cultured for 6 days.

The 8–12% separated gel and 5% concentrated gel were used for gel formation, and wells were created before the gels formed. A total of 60 mg of total protein per hole was injected to obtain an SDS-PAGE gel, and then, the SDS-PAGE gel was electrophoresed for 2 h. The PVDF membrane was soaked in methanol for 20 s and then transferred to tris-glycine transfer buffer (containing 5% methanol) for at least 5 min to attain equilibrium. The SDS-PAGE gel was equilibrated using tris-glycine transfer buffer for at least 30 min. Under cooling conditions, the protein was transferred from the SDS-PAGE gel to the PVDF membrane. Then, the PVDF membrane was added to T-TBS (tris-buffered saline, pH 7.4, containing 0.1% Tween20 and 5% skim milk powder or BSA), equilibrated for 1 h at room temperature and rinsed with T-TBS 3 times for 5 min each. After a primary antibody was added, the PVDF membrane was incubated overnight at 4 °C. Then, a secondary antibody was added. The transfer film was incubated with 1 mL ECL working solution at room temperature for 1 min and then sealed using clingfilm for X-ray development. Image J 1.53e was used to analyze the optical density values of the bands, and each band was repeated 3 times.

## 3. Results and Discussion

### 3.1. Physical Properties and Chemical Structure of WPU Scaffolds

The chemical structures of PU5050, PU7525 and PU9010 were observed using surface infrared spectroscopy (ATR-FTIR). A strong, sharp peak can be observed on the curve at around 1700 cm^−1^, which is attributed to carbonyl C=O in the carbamate; moreover, a N-H vibration peak on the polyurethane can be observed at 3100–3500 cm^−1^, which means that the WPU was successfully synthesized (Figure 1b). The compositions of the materials are similar, with the only difference being the LA/GA ratio in the PLGA, so their infrared curves are very similar. However, a difference of between 1160 and 1190 cm^−1^ can still be found. In the -C (O) -O-CH_2_—structure, the characteristic peak of the C-O- bond occurs at around 1160 cm^−1^, and for the -C (O) -O-CH—structure, the characteristic peak of the C-O- bond moves to around 1190 cm^−1^. In other words, as the proportion of LA in the PLGA increases, the proportion of tertiary carbon in the WPU molecular chain increases, causing the peak in the C-O bond to move from 1160 cm^−1^ to 1190 cm^−1^. As shown in Figure 1b, from PU5050 to PU9010, the peak area at 1160 cm^−1^ gradually decreases, while that at 1190 cm^−1^, it gradually increases. This confirms that the PLGA was successfully synthesized into the molecular chain of polyurethane [19].

All the scaffolds are in a wet state in vivo. Therefore, the performance of the scaffolds in the wet state is crucial. Water absorption can cause changes in the mechanical properties of scaffolds from the dry state to the wet state. In Figure 1a, PU7525 has the highest water absorption at 388%, and the water absorption of PU5050 and PU9010 is 368% and 304%, respectively. LA is more hydrophobic than GA, with an increase in LA, it was increasingly difficult for water molecules to enter WPU scaffolds, and the water absorption of the scaffolds gradually decreases. It is indisputable that the water absorption of the sample is affected not only by the hydrophilicity of the chain segment but also by its crystallinity and other aggregation structures [64]. With the increase in hydrophilicity, the water absorption rate of materials becomes higher, while with the increase in crystallinity, the water absorption rate of materials decreases. These two factors jointly determine the water absorption rate of materials. PU7525 has the highest water absorption rate, which may be due to interference with crystallinity caused by the change in the LA ratio, allowing water to enter the molecular chain more easily. High water absorption can quickly infiltrate the WPU scaffolds with body fluids and facilitate the transportation of metabolic waste and nutrients, and an appropriate modulus could accelerate the attachment of cells.

Matching the mechanical properties of WPU scaffolds with tissues is also a factor that must be considered. Scaffolds that are too soft will not provide adequate mechanical support for regenerated tissue, while scaffolds that are too stiff will cause secondary damage to injured sites. Thus, the mechanical properties of scaffolds were characterized in both the dry state and the wet state to verify whether the WPU scaffolds were suitable for use as implant materials. As shown in Figure 1c,d, the compression strength of dry scaffolds increases as LA/GA increases, and the compression strength of PU5050, PU7525, PU9010 were 37 kPa, 42 kPa, 91 kPa, respectively. In particular, at 70% deformation, PU9010 has the highest compressive strength of almost 91 kPa, more than twice the strength of PU5050 (37 kPa). In the wet state, these scaffolds present the same trend as in the dry state, but their values are considerably lower than those in the dry state. Because PLA has higher strength than PGA, increased LA would strengthen WPU scaffolds. As for the modulus, PU9010 has the highest modulus of 87.8 kPa in the dry state, and the moduli of PU7525 and PU 5050 are 40.9 kPa and 35.1 kPa, respectively. Similar to strength, the value of the modulus in the wet state is much smaller than that in the dry state. The modulus of the wet PU9010, PU7525 and PU 5050 scaffolds are 30.2 kPa, 12.9 kPa and 6.4 kPa, respectively, which match the modulus in neural tissue (2–200 kPa) [68]. High water absorption may cause the strength and modulus to decrease considerably. If water were to enter a gap in the molecular chain, the intermolecular force would be weakened, and the modulus and strength would decrease considerably. All these conditions make WPU scaffolds the ideal material for nerve tissue repair.

### 3.2. Degradation Properties and Cytotoxicity of the Degradation Solution in WPU Scaffolds

Given that scaffolds are used as implanting materials, there are many concerns about their degradation properties. The degradation rate of scaffolds is expected to match the regeneration rate of tissues, so an adjustable degradation rate is a necessary condition for scaffolds. According to previous studies, LA/GA has a great influence on the degradation rate of PLGAs; as LA/GA increases, the degradation rate decreases [64]. This is because LA is more hydrophobic than GA. It can be seen from Appendix A that the WCA of the three WPU films was between 60 and 70°, and with the increase in the LA proportion, the WCA increases, which means that the hydrophilicity of waterborne polyurethane has slightly deteriorated. To adjust the degradation rate of scaffolds, as in the case for the PLGA, PLGAs with three LA/GA ratios were added into the soft segment of the WPU. The result was as expected, with the degradation rates of WPUs containing three different ratios of LA/GA showing the same trend as those in PLGAs (Figure 2a). The degradation rate of PU5050 is the fastest, and its mass retention rate is less than 80% over 10 weeks. PU7525 has a moderate degradation rate, which retained 83% mass at 10 weeks. The degradation of PU9010 is the slowest, and the mass retention rate of PU9010 is 86%. The degradation rate of WPU is considerably affected by LA/GA; thus, regulation of the degradation rate of the WPU scaffolds was realized. The higher the proportion of GA in the PLGAs, the easier it was for water to enter the gap in the molecular chain and attack the polymer group, promoting hydrolysis of the scaffold. Meanwhile, PGA is a highly crystalline polymer. If it contains too much GA, it may lead to crystallization in the PLGA and, in turn, hinder the hydrolysis of the polymer. These are the reasons why PU5050 exhibited the fastest degradation [69].

The pH value also played an irreplaceable role in the process of degradation. As shown in Table 2, the decrease in pH is most obvious in the first week of hydrolysis because autocatalysis led to rapid degradation in the scaffolds at the initial stage. Parallel to the weight retention rate, the pH of the PU5050 scaffold also decreases the fastest, reaching almost 6.1 (which represents a relatively acidic environment) at the tenth week. Correspondingly, the pH values of PU7525 and PU9010 decrease slowly, but the pH of PU9010 was lower than that of PU7525. Although the hydrolysis process of PU9010 was slower, PU9010 contained a higher proportion of LA, and the acidity of LA was higher than that of GA. This resulted in the release of more LA molecules during the hydrolysis process of PU9010, making the pH of the degradation solution for PU9010 lower than that for PU7525.

To meticulously analyze the in vitro degradation process, the samples were taken out at the first, fourth and tenth weeks of in vitro hydrolysis and observed using SEM. In Figure 2c, the images showing all sections of the scaffolds at the initial stages presented a cellular structure which expanded the inside surface area of the scaffolds and provided more sites for the cells to adhere. More subtly, all the scaffolds showed smooth inner walls and small cracks at the edges, which increased the surface area of the scaffolds even further. At the first week of degradation, it was observed that all scaffolds lost much of their mass in the first week (Figure 2a); this mainly resulted from the precipitation of low molecular weight polymers in the waterborne polyurethane and mild degradation, and it did not have much impact on the structure of the scaffolds. As observed in Appendix A, the SEM images of the scaffolds that were degraded for one week are not significantly different from the initial SEM images of the scaffolds (Figure 2c). Furthermore, the PU9010 scaffolds have smooth edges while keeping smooth walls, whereas the PU5050 and PU7525 have jagged edges despite keeping the walls smooth, which is indicative of slight hydrolysis of the scaffolds. As the fourth week of the degradation process approached, the inner walls of all scaffolds showed different degrees of fracture and cracking, among which PU5050 had the highest degree of roughness, PU7525 had fewer fractures and PU9010 had the lowest degree of degradation and still maintained a relatively smooth inner wall. These results were consistent with those for the mass retention ratio. When the hydrolysis process reached the tenth week, there was further degradation in the PU5050, PU7525 and PU9010 scaffolds, and the fractures and cracks continued to develop. More folds appeared on the inner wall surface of the WPU scaffolds, especially on the surface of PU5050 scaffolds, where horizontal grooves appeared, indicating that the effects of hydrolysis had been significant. The degree of corrosion on the surface of PU7525 was still medium, and the degree of hydrolysis of PU9010 was the lightest, but the surface was no longer smooth and exhibited more lines. These images indicate that hydrolysis of the scaffolds occurred over time. Upon combining the mass retention curve, pH value curve and the WPU scaffold SEM images, it is evident that the WPU scaffolds degraded more slowly as the LA/GA ratio increased. It can be asserted that the degradation rate of the WPU scaffolds was adjusted successfully. Furthermore, the scaffold degradation rates match the repair rates of different tissues; the pH value was also regulated during degradation, thus affecting the inflammatory response of the damaged tissue.

In order to confirm that the acidity during degradation process had no effect on the activity of cells, the diluent degradation products of the three scaffolds were reduced 10 times, 100 times and 1000 times, respectively, at the fourth and tenth weeks; then, L929 cells were cultured using these diluted culture solutions. Finally, the viability of the cells was characterized using the MTT method. The viability of all the L929 cells cultured using the diluent degradation solutions of these scaffolds was higher than 80% (Figure 2c), indicating that the diluent degradation solutions of these scaffolds possessed no obvious cytotoxicity and would not have adverse effects on cell growth and proliferation. The customizable degradation rate and excellent biocompatibility suggest that these scaffolds are ideal candidates for nerve tissue engineering.

### 3.3. Effects of the LA/GA Ratio in WPU Scaffolds on the Inflammatory Response In Vitro

Implant materials could cause severe inflammation due to immune rejection, and the inflammatory response is critical to the growth of nerve cells. Thus, the inflammation response is a key factor that must be considered in the development of biomaterials for nerve tissue repair. Inflammation in damaged nerve tissue is primarily adjusted by activated microglia; therefore, the microglial BV2 cell line (cell line that plays an irreplaceable role in the immune response in the brain) was chosen as a representative for primary microglia. BV2 cells are similar to macrophages and have two phenotypes: the proinflammatory phenotype (M1) (which expresses the iNOS protein and secretes TNF-α and other factors) and the anti-inflammatory phenotype (M2) (which expresses the Arg1 protein and secretes IL-10, IL-4 and other factors). In order to explore how LA/GA in WPU affects the immune microenvironment at the repaired site, we seeded BV2 cells on WPU scaffolds.

NO is one of the signature secretions of BV2 cells with the proinflammatory M1 phenotype. The NO released from BV2 cells on scaffolds and TCPs, with or without LPS stimulation, was measured using a NO kit. As can be seen in Figure 3d, in the absence of LPS stimulation (LPS is often used to simulate the inflammatory response at the damaged site), the NO secreted from BV2 cells on the WPU scaffolds was comparable to that in the control group (TCP). Though the level of NO secretion by BV2 cells cultured on the PU7525 scaffold (0.083) was higher than on the other scaffolds, it was in the same order of magnitude as the NO secreted by BV2 cells on TCP (0.058). In this stimulation, the WPU scaffolds did not stimulate BV2 cells to produce more NO, suggesting that WPU scaffolds themselves did not cause a strong inflammatory response. When stimulated by LPS, the BV2 cells cultured on scaffolds showed little change in NO secretion compared to those without LPS stimulation, and the BV2 cells on WPU scaffolds even secreted less NO. Among the three scaffolds, the secretion of NO by BV2 cells was still the highest on PU7525 (0.072) and was much lower than in the TCP group (0.13); perhaps related to being the most hydrophilic, the BV2 cells on PU5050 emitted the lowest NO (0.054). This revealed a significant difference between NO release with and without LPS stimulation. These results show that none of the scaffolds promoted the differentiation of BV2 cells into the proinflammatory M1 phenotype by themselves and that even WPU scaffolds can help alleviate external inflammatory stimulation when stimulated by LPS. 

To further understand the effect of different ratios of LA/GA in scaffolds on BV2 cells differentiation, the supernatant of the cell culture was tested using ELISA, and the cells cultured on the scaffolds were tested using WB analysis. The secretion levels of TNF-α (pro-inflammatory phenotype marker factor), IL-10 (anti-inflammatory phenotype marker factor) and IL-10/TNF-α (an index often used to measure inflammatory responses [70]) without LPS are listed in Figure 3a–c. TNF-α (79.8 pg/mL) was highest on the PU7525 scaffolds, while the BV2 cells secreted less TNF-α on the PU9010 and PU5050 scaffolds (40.8 and 39.5 pg/mL, respectively), similar to the TCP sample (24.8 pg/mL). Simultaneously, the secretion of IL-10 from BV2 cells was lowest on the PU7525 scaffolds (41.6 pg/mL), and the BV2 cells emitted slightly higher levels of IL-10 on PU9010 and PU7525 (48.3 and 48.9 pg/mL, respectively). The secretion of IL-10 by the BV2 cells was higher on WPU scaffolds than in the whole TCP group, suggesting that WPU scaffolds containing PLGAs can better induce BV2 cells to differentiate into the M2 phenotype. In Figure 3c, the IL-10/TNF-α ratio of PU7525 (0.596) was lower than in the TCP sample (1.522), which indicates that PU7525 caused a relatively severe inflammatory response. However, the IL-10/TNF-α ratio of PU9010 and PU5050 was similar to that in the control sample, which means these materials did not promote the formation of a significant inflammatory environment. Further, the results of the WB test also show the same trend (Figure 3e). As for iNOS, a proinflammatory marker protein, the protein secretion levels for all three scaffolds were not significantly different from those in the control sample. Meanwhile the protein level of Arg-1, an anti-inflammatory marker protein, on the WPU scaffolds was significantly higher than that in the TCP sample. Overall, these results suggest that WPU scaffolds did not promote the differentiation of BV2 cells into the proinflammatory M1 phenotype. When stimulated by LPS, the WPU scaffolds were shown to possess strong anti-inflammatory properties (Figure 3a–c). With an increase in LA/GA, the TNF-α secretion levels from the BV2 cells gradually increased but were all lower than in the TCP sample (1366.9 pg/mL). This may be because with the increase in LA, the hydrophilicity of the material became worse and worse, leading to the gradual decline in the affinity of the scaffold for BV2 cells. On the one hand, the secretion of IL-10 from BV2 cells gradually increased from 127.2 pg/mL in PU5050 to 133.8 pg/mL in PU7525 and 138 pg/mL in PU9010, respectively. The expression levels of IL-10 were all higher than in the control sample (72.1 pg/mL), and in PU9010, they were nearly twice as high as in the control group. This means that BV2 cells cultured on WPU scaffolds were more likely to differentiate into the anti-inflammatory M2 phenotype. As for IL-10/ TNF-α, the differences between the WPU scaffolds and control group were more obvious. The ratio in the WPU scaffolds was higher than in the TCP sample, the lowest ratio in the PU9010 scaffolds (0.16) was three times higher than that in the control group (0.053), and even the largest ratio in the PU5050 scaffolds (3.64) was nearly 68 times greater than that in the control group. This indicated that these three WPU stents can effectively reduce the inflammatory response at the implantation site. On the other hand, the results of the iNOS and Arg-1 WB tests were highly consistent with those of ELISA (Figure 3f). As the LA/GA ratio increased, the expression of pro-inflammatory iNOS and anti-inflammatory Arg1 increased. Therefore, the WPU scaffolds’ anti-inflammatory properties were confirmed in the presence of LPS. In short, not only did WPU scaffolds not cause inflammatory responses on their own, but they could also alleviate inflammatory responses in injured tissues. Thus, the effect of LA/GA on the immune response in BV2 cells was clearly determined. The differences in the mechanical and hydrophilicity of WPU scaffolds may lead to varying differentiation directions in BV2 cells. In general, compared to the TCP group, without LPS, WPU scaffolds did not induce BV2 cells to excessively differentiate into the pro-inflammatory M1 phenotype. When stimulated by LPS, WPU scaffolds can prevent BV2 cells from differentiating into the pro-inflammatory M1 phenotype. With decreasing LA/GA, the anti-inflammatory effect of WPU scaffolds becomes better and better because better hydrophilicity and a lower modulus can reduce the inflammatory response of tissues. The result of the inflammatory responses of BV2 cells reinforces the belief that WPU scaffolds are excellent candidates for neural tissue repair.

### 3.4. Seeding PC12 Cells on WPU Scaffolds

A suitable wet state modulus, high water absorption, honeycomb pore structure and, most importantly, preventing BV2 cells from transforming into the pro-inflammatory M1 phenotype, are clues that indicate WPU scaffolds are ideal candidate for repairing nerve tissue. To verify whether the scaffolds can promote the growth, proliferation and differentiation of nerve cells in vitro, as the most-used extracorporeal nerve cell model, PC12 cells were implanted on the scaffolds. The PC12 cells grew into WPU scaffolds (Figure 4a), which confirmed that scaffolds can help in the adhesion and growth of PC12 cells. In terms of cell morphology, the PC12 cells cultured on TCP were round with a clear boundary, which occurred because they were undifferentiated [71]. However, compared with the TCP group, the PC12 cells grown on WPU scaffolds either grew into clusters or grew alone. Although it was difficult to observe their synapses after differentiation, the cell morphology of PC12 cells was no longer round, and their cell boundaries became blurred. Moreover, some neurites could be vaguely identified from the zoomed-in confocal microscope pictures. In order to further confirm whether the PC12 cells had differentiated, their secretion of GAP-43 was measured using WB analysis to corroborate the confocal microscope pictures. As a secretion marking the differentiation of PC12 cells, the amount of GAP-43 can be used to measure the repairing effect in nerve tissue. Figure 4b shows that the secretion of GAP-43 from PC12 cells on all WPU scaffolds was much higher than that in the TCP group. The PC12 cells on PU5050 scaffolds secreted the most GAP-43 (2.01), with their level nearly six times that in the TCP group. Even the lowest secretion on PU7525 (1.06) was nearly three times higher than that in the control group (0.38). The fact that the lowest secretion of Gap-43 occurred in PU7525 may be mainly due to its poor inflammation reduction properties. It has been reported that with an increase in M1/M2 in BV2 cells, the ability of PC12 to produce neurites, and the neurite length, decreased [71]. This is why IL-10 secretion without LPS stimulation (Figure 3a) maintained the same trend as the secretion of GAP-43 (Figure 4c). Combined with confocal microscope images, WPU scaffolds were proven to have the ability to promote PC12 cell differentiation (with even the worst performing PU7525 performing much better than the TCP control group). Moreover, WPU scaffolds were shown to have great potential for repairing damaged nerve tissue. Although the modulus of the WPU scaffold is very suitable for nerve tissue repair and has a significant effect on the differentiation type of BV2 microglia cells, and even the results of PC12 culture in vitro show that the WPU scaffold is very suitable for nerve tissue repair, subsequent verification such as animal experiments is necessary. At the same time, the modulus of WPU scaffolds reaches a wide adjustment range (from 6.4 kPa to 30.2 kPa), but WPU scaffolds are not suitable for all soft tissue repair, especially for muscles, ligaments and other tissues that require high mechanical properties.

## 4. Conclusions

In this study, WPU scaffolds were prepared using PLGAs with different ratios of LA/GA in place of PCL. That is, under the condition that the PLGA occupied a certain proportion of the soft segment, PLGAs with different LA/GA ratios were selected to obtain different polyurethane scaffolds (PU5050, PU7525 and PU9010). The three WPU scaffolds had similar water absorption levels, high porosity and similar pore structure. Due to the differences in PLGA content, the three scaffolds showed different mechanical and degradation performance, which qualified them for repairing nerve tissue. Especially, the wet state modulus of WPU scaffolds were between 6.4 kpa and 30.2 kpa, which perfectly met the requirements of nerve tissue repair. In terms of their anti-inflammatory effects, upon culturing BV2 cells on the scaffolds with or without LPS stimulation, the scaffolds, particularly the PU5050, did not cause a strong inflammatory response and were able to relieve inflammation caused by external stimuli; even the IL-10/TNF-α of PU5050 (3.64) reached 69 times that of the control group (0.053). In addition, using confocal microscopy and GAP-43 protein expression, it was verified that the three scaffolds can support the growth, proliferation and differentiation of PC12 cells. Thus, it can be stated that such scaffolds have potential for application in neural tissue repair. Although the WPU stent has shown promising applications in nerve tissue repair, further animal experiment validation and the tissue universality of the WPU scaffolds need to be verified.

## Data Availability

Not applicable.

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
