# Peer review of "In Vitro Effects of Waterborne Polyurethane 3D Scaffolds Containing Poly(lactic-co-glycolic acid)s of Different Lactic Acid/Glycolic Acid Ratios on the Inflammatory Response"

_polymers, 2023, doi:10.3390/polym15071786_

Round 1

Reviewer 1 Report

In the manuscript authors reported the waterborne polyurethane (WPU) scaffolds using different ratios of LA/GA PLGA (PU5050, PU7525 and PU9010). All these three scaffolds exhibited distinct mechanical and degradation performances but comparable porosity, pore size, and water absorption. Furthermore, the BV2 cells were cultured on the scaffolds with or without LPS stimulation and there was no strong inflammatory response observed. Moreover, the authors claimed that these WPU support growth, proliferation, and differentiation of PC12 cells and such scaffold has a promising potential application in neural tissue repair.

Therefore, the manuscript addresses the potential interest of the study.

However, the provided data sets to support the conclusion is not sufficient and thus it seems premature to proceed with the manuscript based on the current results.

1.       The authors should add water contact angle for all these three WPUs.

2.       The authors should add electron microscopy images for 1st week of hydrolyzing process for WPU scaffolds.

3.       In figure 4, the data for LPS and without LPS should be merged to see the actual comparison. The authors should also add significance of data after comparison.

4.       The scales in figure 5a are not correct.

5.       The authors should add a comment on outcome of PU5050, PU7525 and PU9010 scaffolds when co-cultured with BV2 microglia regarding whether they promote the differentiation of BV2 into an anti-inflammatory M2 phenotype or pro-inflammatory M1 phenotype.

Reviewer 2 Report

This manuscript studies the performance of waterborne polyurethane 3D scaffolds with different lactic acid/glycolic acid ratios. The subject is interesting and the manuscript has been well written. However, some revisions should be addressed before the final decision.

1. Please avoid using acronyms in the title.

2. In line 15, WPU has been used; however, its acronym should be introduced before. For instance, in line 12.

3. Please avoid using the lumped references such as [3-5], [12-15], [20-22], and etc. Each sentence should have one reference and the references should not be presented without justification. It is better to break down the lumped references.

4. Only the acronyms should be presented that they are used several times in the manuscript. For instance, VOC acronym has been introduced in line 65 as the abbreviation of volatile organic compound; however, it is not used in the manuscript. Please pay attention to using the acronyms.

5. Please provide numbers for equations such as line 151, line 162, line 179.

6. Please provide supporting references in section two where they are required. For instance, the porosity formula needs supporting reference.

7. Please provide the y-axis titles in Figure 1-b.

8. The standard deviations of the results are very large in most of cases. These wide error bars endanger the conclusions that there are significant differences between the results. If a statistical analysis is conducted such as 2-sample t-test, it will be revealed that the results do not have significant differences.

9. The conclusions section is good but it has been repeated.

Reviewer 3 Report

  1. Graphical abstract is submitted during submission in MDPI system, not in the manuscript.
  2. Line 14, LA/GA, please state the stand of its abbreviation first.
  3. Line 17, what is BV2? Mention the stand for it first.
  4. Line 18, WPU? Please state Waterborne polyurethane first before use the abbreviations.
  5. Quantitative results need to be added in the abstract section.
  6. Please give a "take-home" message as the conclusion of your abstract.
  7. Rearrange the keywords so that they are in alphabetical order.
  8. Line 26-29, the authors explain the role of bone scaffold in tissue engineering. Please giving additional reference for the explanation to support it. Suggested reference needs to incorporated as follow: The Effect of Tortuosity on Permeability of Porous Scaffold. Biomedicines 2023, 11, 427. https://doi.org/10.3390/biomedicines11020427
  9. What is the novel bought by the authors in the current submission? Its works have been widely discussed in the past. Nothing something really new in the present form. The lack of a novel seems to make the present submission like to replication/modified work. The authors need to detail their novelty in the introduction section. It is a major concern for rejecting this paper.
  10. In order to highlight the gaps in the literature that the most recent literature aims to fill, it is crucial to review the benefits, novelty, and limitations of earlier studies in the introduction.
  11. It is suggested to the authors to make the objective of the present work become more clear to understand.
  12. Encouraging to the authors to provide an additional figure in the introduction section for increasing the quality of the present submission.
  13. Line 127, what is urgency of equation in table 1? I think it should be separated into different equation, not combined as a caption in Table 1.
  14. Line 280 in Figure 1, I do not found any something different and make the figure become important since the trends is similar without significant changes.
  15. Line 390 for figure 3 (b) would be recommended to separated from group of figure b into different independent as a table.
  16. Line 470-495, I do not get the importance at this section. Would be better to delete it?

Round 2

Reviewer 1 Report

I recommend this for publication.

Author Response

Thank you for your recommendation

Reviewer 3 Report

Other comments is given as the response:

1.      To help the reader grasp the study's workflow more easily, the authors could include more visuals to the materials and methods section in the form of figures rather than sticking with the text that now predominates.

2.      Given that the current form is inappropriate, the authors must address the basis for patient selection. Is any standard, procedure, or protocol used?  The included patients are extremely small and heterogeneous, and there is no true group control. As it is one of the main problems with the current submissions and the justification for suggesting a rejection, the reviewer kindly ask that you take this matter seriously.

3.      More detail in tools information such as manufacturer, country, and specification needs to be stated.

4.      The paper needs to provide critical information on the error and tolerance of the experimental equipment utilized in this work. Due to the disparate outcomes of subsequent research by other researchers, it would make for an insightful conversation.

5.      A comparative assessment with similar previous research is required.

6.      Since the present study performs In Vitro investigation, rationalization is needed, why not performed in vivo or in silico? Also, the reviewer encouraged the authors to explain potential study adopting in silico/computational simulation study since it bring several advantages compared to In vitro such as lower cost and faster results. In silico would become preliminary step study then following into more advances in vitro/in vivo study. Please refer to the reference as follows: In Silico Contact Pressure of Metal-on-Metal Total Hip Implant with Different Materials Subjected to Gait Loading. Metals (Basel). 2022, 12, 1241. https://doi.org/10.3390/met12081241

7.      Overall, the discussion in the present article is extremely poor. The Authors must extend their discussion and make a comprehensive explanation.

8.      The present article's limitations should be added before moving on to the conclusion section.

9.      Add more detail to the conclusion by structuring it as a paragraph rather than in point-by-point present form.

10.   In the conclusion section, further research must be discussed.

11.   In the entire manuscript, the authors occasionally constructed paragraphs with just one or two phrases, which made the explanation difficult to understand. To make their explanation a full paragraph, the authors should expand it. It is advised to use at least three sentences in a paragraph, with the primary sentence coming first and the supporting sentences coming after.

12.   The authors should give additional references from the five-years back. MDPI reference is strongly recommended.

13.   The authors are encouraged to reduce their self-citation.
